# Phenotypic Correlation between Guard Hair and Down Hair in Chinese Alashan Left Banner White Cashmere Goat: A Preliminary Study

**DOI:** 10.3390/ani13081295

**Published:** 2023-04-10

**Authors:** Stefano Pallotti, Dario Pediconi, Alessandro Valbonesi, Carlo Renieri, Sun Haizhou, Zhou Junwen, Marco Antonini

**Affiliations:** 1School of Biosciences and Veterinary Medicine, University of Camerino, Via Gentile III Da Varano s/n, 62032 Camerino, Italy; 2School of Pharmacy, University of Camerino, Via Madonna delle Carceri, 9, 62032 Camerino, Italy; 3Inner Mongolia Academy of Agricultural and Animal Husbandry Sciences, Zhaoju Road No. 22, Yuquan District, Hohhot 010031, China; 4Alashan League Institute of Animal Husbandry Research, Erlute West Road, Bayanhaote, Bayanhaote 750399, China; 5Italian National Agency for New Technologies, Energy and Sustainable Development Dpt. Casaccia, Via Anguillarese 301, 00123 Galeria, Italy

**Keywords:** cashmere, Alashan, goat, guard hair, hair length

## Abstract

**Simple Summary:**

The cashmere goat’s double coat is characterized by long and coarse guard hair, which provides mechanical protection, and short and fine down hair, the luxury cashmere fiber, which provides thermal protection to the animal. The main commercial attributes of cashmere fiber are its length and fineness and the coarse guard hair is usually discarded by the textile industry. The aim of this study was to describe the phenotypic correlation between the characteristics of the coarse guard hair and the other fiber.

**Abstract:**

In cashmere production studies, few trials have considered the guard hair features and their correlation with down fiber attributes. In this preliminary work, early observations on 158 one year old Chinese Alashan Left Banner White Cashmere goats were carried out. The aim was to describe the phenotypic correlation between the guard hair length and other fiber traits. The guard hair length was positively correlated with guard hair diameter and the down fiber length. Negative correlations were found between guard hair length and the coefficient of variation of guard hair diameter, between the guard hair diameter and its coefficient of variation, and between the down fiber diameter and the coefficient of variation of down fiber diameter. The body weight at first combing was not correlated with any of the other traits.

## 1. Introduction

The cashmere goat’s double coat is characterized by long, medullated and coarse guard hair, which provides mechanical protection, and short and fine down hair, the cashmere fiber, which provides thermal protection to the animal [1]. The US Wool Products Labelling Act defines cashmere as the “fine (dehaired) undercoat fiber produced by a cashmere goat, with an average fiber diameter not exceeding 19 μm, and with no more than 3% of the fibers (by weight) having an average diameter that exceeds 30 μm, and a CV of diameter not exceeding 24%” [2].

This luxury fleece is one of the warmest animal fibers and is mainly used to produce high-quality and costly clothes [3,4]. China and Mongolia produce the 90% of world cashmere [5,6,7]. The annual Chinese cashmere production was estimated at some 15–18 million kg [8,9] of which the ~45% were produced in Inner Mongolia [9].

The Inner Mongolian cashmere is famous worldwide for its fineness, in particular, the fiber produced by the Alashan Left Banner White Cashmere goats (ALBWCG), is known to be the finest Chinese cashmere [10,11]. Above all, the most precious fiber produced by the ALBWCG, known as “Baby Cashmere”, is harvested from the kids only once, at 5–6 months of age, to collect small batches of about 30 g of fiber 1 μm finer than regular cashmere [12,13].

During the past years, to increase the down weight and the live body weight of the animal, the ALBWCG population was subjected to genetics admixture with the Liaoning and the Inner Mongolian Arbas cashmere goats causing a loss in the fineness and homogeneity of the fiber [11]. In light of this, a genetic improvement program was started in Alashan (Inner Mongolia, China) in 2009 in order to enhance the quality of the ALBWCG fiber. The project is coordinated by the Station for Livestock Improvement and supported by the Chinese Agricultural University of Jilin, the Italian University of Camerino, the Italian National Agency for New Technologies, Energy and Sustainable Economic Development (ENEA) and the Loro Piana S.p.A. textile industry [12]. In cashmere production, most studies have focused on the main down fiber commercial attributes [14,15,16]. Currently, relatively little information has been produced on the cashmere guard hair features and the majority of these studies did not explore the correlation between the guard hair and other quality attributes of commercial cashmere [12,13,17,18,19,20,21,22]. The lack of interest in the study of the guard hair is mainly attributable to it being a low-value product, and the cost and time involved in measuring the guard hair diameter. This thick fiber in fact, is discarded as too coarse for textile use, or possibly used to make brushes and interlinings [2].

However, our early observations on the upper coat of the ALBWCG suggest the guard hair length shows phenotypic correlation with the down fiber length. In view of this, the present paper aims to disseminate data from our measurements on guard hair fiber and its relationship with other cashmere traits in ALBWCG.

## 2. Materials and Methods

### 2.1. Animals

One hundred and fifty-eight one year old goats (71 males and 87 females) born between January and May 2020 were selected at random from the herd located in the Mingyuan-Loro Piana Station for Livestock Improvement in Alashan, Left Banner, a semi-desert steppe area of Inner Mongolia (China). The original sampling herd at the Station counted 750 animals fed with a standardized diet during the cold season. The kids remained with their mothers until weaning at three months of age.

### 2.2. Measurements

Fleece sampling was performed from May to June 2021, during the first combing when the animals were about one year old. As the live body weight has been shown to be the main driver in fiber diameter for several breeds including cashmere [23], the live body weights at first combing were recorded after weighing each animal with a balance. Fiber was cut to skin level in a 10 cm^2^ patch on the right mid-side of the body (behind the scapula). Fiber samples were then shipped to the School of Bioscience and Veterinary Medicine, University of Camerino (Italy) for the measurements. The whole sample of raw cashmere fiber was separated manually into down hair and guard hair then washed in ether solution to remove grease and contaminants such as soil. The maximum lengths of unstraightened down hair and guard hair were determined to the nearest 1 mm by laying the undisturbed sample flat.

Down fiber diameter and guard hair diameter were measured with an optical fiber diameter analyzer (OFDA 100). The mean, standard deviation, and coefficient of variation were computed for the samples. A minimum of 4000 snippets of white fiber were measured for each sample. Snippets were prepared according to the standard test method for diameter of wool and other animal fibers using an optical fiber diameter analyzer proposed by ASTM [24].

### 2.3. Statistical Analysis

The Kruskal–Wallis test was performed to compare the differences between males and females for all the traits recorded in the trial. Spearman’s Rho (*r_s_*) test was used to measure the strength of correlation between all the variables studied. All the statistical analyses were performed using IBM SPSS Statistics 21 software [25].

## 3. Results

The results of the statistical analysis are showed in the Table 1. All the variables were normally distributed.

Our measurements showed that ALBWCG reached a mean live body weight at first combing (about one year old) of 22.4 kg producing cashmere with a mean down diameter of 13.8 µm, a CVDFD (coefficient of variation of down hair diameter) of 25.37 and a mean maximum crimped cashmere staple length of 5.66. The guard hair showed a mean diameter of 47.3 µm, a coefficient of variation of down guard hair diameter 31.35 and a maximum length of 15.2 cm. The Kruskal–Wallis test showed significant differences between males and females only for the mean of down fiber diameter (*p* value = 0.001) and CVDFD (*p* value = 0.021) (Table 2). Males showed a lower down fiber diameter compared to the females but a higher CVDFD (Table 3). No significant differences were found for the mean of the other traits analyzed considering sex as factor.

As showed in the Table 4, significant positive correlation has been found between maximum guard hair length and maximum crimped cashmere staple length (*r* = 0.410) while significant and negative correlation were found between maximum guard hair length and the coefficient of variation of guard hair diameter (*r* = −0.427). Significant negative correlations were observed between the guard hair diameter and it’s coefficient of variation (*r* = −0.398) and between the down fiber diameter and the CVDFD (*r* = −0.291). 

## 4. Discussion

Among all the cashmere attributes studied, statistical analysis showed differences between male and female only in the mean down fiber diameter and CVDFD. As showed in the Table 3 in fact, males produce a slightly finer fiber (about 0.5 µm). On the other side the cashmere produced by the females was more homogeneous due to its lower CVDFD (−1.29). Higher performance by males in terms of cashmere fineness has been already reported by previous studies on ALBWCG [13].

Our results showed the higher cashmere quality of the ALBWCG in terms of fiber fineness (mean 13.8 µm) compared to other common cashmere breeds such as the Liaoning and the Raeini cashmere goat which had a mean down diameter of about 17.5 µm and 18.0 µm, respectively [26,27]. The maximum crimped cashmere staple length observed in the ALBWCG (5.6 cm) could be appropriate to cashmere industry as a fiber length greater than 3.6 cm is considered ideal for the industrial processing [28].

It is known that dehairing is one of the most crucial steps in cashmere processing, the process efficiency has a high impact on the quality and the price of the knitted fiber. The main fiber features which play a central role in the effectiveness of mechanical dehairing are the hair rigidity (defined mainly by its diameter) and the hair length. In this regard, it is crucial to avoid the overlapping between the short and fine down fiber and the coarse and long guard hair [2,29,30].

The wide differences between the down hair and the guard hair in terms of maximum fiber length and diameter, which were of 5.6 cm and 15.2 cm, and 13.8 µm and 47.3 µm, respectively, should be noted. These preliminary observations suggest that the ALBWCG fleece structure could be suitable for high efficiency in the dehairing process. However, it must be stressed that our measurements were carried out only on a mid-side sample, therefore specific experiments on dehairing efficiency are needed to shed light on this aspect. 

In this study, a greater focus has been put on the measurement of the guard hair parameters to identify any correlation between the upper coat fiber and the others cashmere parameters. In this respect, it is noteworthy that the maximum guard hair length showed strong and significant positive phenotypic correlation of 0.41 with the maximum crimped cashmere staple length, an important economic trait in cashmere production [31]. However, this preliminary phenotypic observation needs to be validated by genetic data.

Similar to our results, no phenotypic correlations were observed by Wang and colleagues [28] (2013) between guard hair length and the down diameter for the Inner Mongolia Cashmere goat.

It must be stressed that the present work has some limitation that must be considered. First, our outcomes were based on observation carried out on the mid-side site of 158 goats, therefore the results have to be corroborated using a larger sample. Our trial explored only phenotypic correlations between different fiber features; therefore, genetics parameters for all the traits considered in the ALBWCG population have to be estimated. Above all, it is important to assess whether long-term selection for guard hair length might affect the down fiber diameter. Trials have to be planned in order to evaluate the impact of the increase of the guard hair length on the combing activity and industrial/manual dehairing process.

Further work is needed to clarify all these issues.

## 5. Conclusions

The present work aimed to describe the structural relationship between the guard hair and the down hair in the ALBWCG. Preliminary results suggest that maximum guard hair length is phenotypically correlated with the maximum crimped cashmere staple length; in contrast, no relationship was found between the maximum guard hair length and the down fiber diameter.

Such considerations are based on preliminary observations. Further studies are needed to implement an efficient cashmere selection plan.

## Figures and Tables

**Table 1 animals-13-01295-t001:** Descriptive statistics for fiber traits in ALBWCG.

Trait	Mean	SD	Min	Max
Live body weight at first combing (kg)	22.4	3.45	16.3	33
Maximum guard hair length (cm)	15.2	4.6	1	28
Guard hair diameter (µm)	47.3	6.29	33.6	65.5
Coefficient of variation of guard hair diameter	31.3	8.38	21.4	58.6
Maximum crimped cashmere staple length (cm)	5.6	1.44	2.9	10.7
Down hair diameter (µm)	13.8	0.77	11.9	15.4
Coefficient of variation of down hair diameter	25.3	2.82	18.7	34.4

**Table 2 animals-13-01295-t002:** Summary table of Kruskal–Wallis test results comparing fiber traits between males and females (significance level of 0.05).

Trait	Significance
Live body weight at first combing (kg)	0.697
Guard hair length (cm)	0.364
Guard hair diameter (µm)	0.601
Coefficient of variation of guard hair diameter	0.727
Down length (cm)	0.218
Down diameter (µm)	0.001
Coefficient of variation of down hair diameter	0.021

**Table 3 animals-13-01295-t003:** Descriptive statistics for down fiber diameter and CVDFD in males and females respectively.

	Male		Female	
Trait	Down Diameter (µm)	CVDFD	Down Diameter (µm)	CVDFD
Mean	13.5	26.0	14.0	24.7
SD	0.70	2.58	0.71	2.9

CVDFD = Coefficient of variation of down fiber diameter.

**Table 4 animals-13-01295-t004:** Spearman’s Rho phenotypic correlations between fiber traits.

Trait	BW	GHL	GHD	CVGHD	DL	DFD
GHL	0.067					
GHD	0.008	0.308 *				
CVGHD	−0.049	−0.427 **	−0.398 **			
DL	0.043	0.410 **	−0.009	0.072		
DFD	−0.024	0.193	0.074	0.028	0.124	
CVDFD	−0.057	−0.063	−0.059	−0.091	−0.198	−0.291 **

BW = Body weight at first combing; GHL = Maximum guard hair length; GHD = Guard hair diameter; CVGHD = Coefficient of variation of guard hair diameter; DL = Maximum crimped cashmere staple length; DFD = Down fiber diameter; CVDFD = Coefficient of variation of down fiber diameter. Significance: * *p* < 0.05; ** *p* < 0.001.

## Data Availability

Data can be shared upon reasonable request.

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
