# Peer review of "Phenotypic Correlation between Guard Hair and Down Hair in Chinese Alashan Left Banner White Cashmere Goat: A Preliminary Study"

_animals, 2023, doi:10.3390/ani13081295_

Round 1

Reviewer 1 Report

There is scientific interest in the development of the fleece of cashmere bearing goats and as the authors state, there are relatively few studies of the physical properties of the guard hair component. From this solid starting point the authors let themselves down by some poor methodology, poor understanding of scientific terms and confused reasoning.

There is a potential report here but it must have a more narrow focus, be very clear about the actual methods used, not be distracted by minor issues and avoid making conclusions about issues which are outside the narrow focus of the preliminary investigation. The discussion needs to focus on the main finding, if any, of the work. The data could potentially show that guard hairs are not useful for predicting cashmere content of a fleece compared with direct measurement of cashmere.

The study has some important limitations which the authors do not seem to appreciate. They have not controlled for variation in birth and sampling dates which are highly likely to lead to differences in the period of growth of guard hair of up to 6 months and therefore confound the measurement of guard hair length. One of the cited references shows this type of result but is not used in the discussion.

The authors also confuse themselves and the reader by not providing a clear explanation of the cashmere combing harvesting whereby most of the guard hair remains on the goat after the combing.

There is also confusion by suggesting that an indirect method of assessing cashmere should involve measuring guard hair length, when it would be better to instead spend your time directly measuring cashmere length, especially given the low correlation between hair length and cashmere length in the present study (0.41), which explained only 16% of the variance in cashmere length. This result might indicate the opposite conclusion to the one presently made in the manuscript.

There are already quite a few published papers on the usefulness of measuring cashmere length in predicting total cashmere production, but mid-side sampling is not necessarily a reliable method of predicting total cashmere production from cashmere goats (see later comments).

Specific comments

L68 replace few with little to read … little information has been …

L71 … to it being a low value product.

L71, a main contributor to the lack of reporting is most likely the cost and time involved in measuring the hair diameter.

L72, … or possibly used to …

L80 Do not start a sentence with a number. Remove the full stop after the number.

L92. Rewrite as awkward and potentially misleading expression.

Reference 27 is outdated and not suitable for cashmere and should be removed. Since 1968 more thorough studies with Merino sheep in Australia have shown that the mid-side site does not estimate the fibre diameter of wool tops as well as grid sampling of the fleece. McGregor 1994 also demonstrated systemic bias of mid-side sampling in cashmere goats for both cashmere content and also differences for cashmere fibre diameter. The real issue is that mid-side sampling provides reasonable ranking of goats if it is ranking that you desire but not necessarily predictions of cashmere fibre. This approximation will consequently modify any result and claims made about your subsequent results. What McGregor 1994 reported in relation to the mid-side site systematically overestimating the cashmere yield will impact upon the usefulness of guard hair measurements.

McGregor, B.A. (1994). Measuring cashmere content and quality of fleeces using whole fleece and midside samples and the influence of nutrition on the test method. Proceedings of the Australian Society of Animal Production 20: 186-189. This paper was easily found by Google on Research Gate.

L96-97. The sample would not be undisturbed if it had been washed in ether. Is the sequence of events correct?

L96. If the fibre sample was undisturbed then the length measurements would be the unstraightened lengths. Given that cashmere has natural waves which are called crimp, the actual cashmere length would be longer than those measured.

L96 and elsewhere. There seems to be something missing as the manuscript presents data for both the CV of guard hair length (lines 138, 140) and CV of cashmere length based on only one measurement per sample?

L98. You have not defined what you mean by staple weight or really how you selected a staple for weighing.

L99. To what precision were the weights recorded, 0.01, 0.001 g?

L100-101. There is confusion here, a proportion would be say 0.45, while the percentage would appear as 45%. Clarify this sentence.

L104. How were the snippets prepared?

Are you sure that 4000 snippets were measured on the hair samples without repeat measurements of the same field of view? As table 1 shows, some samples only had 0.02 g of hair to measure.

L109, replace carried on with undertaken.

Results

To ensure that the reader is clear about the measurements please replace the following descriptions throughout the manuscript:

Guard hair length replace with Maximum guard hair length (see line 96);

Down length replace with Maximum crimped cashmere staple length.

Table 1 and elsewhere in the manuscript.

a)     Please note that as the OFDA 100 only measures to the nearest 1 micrometre, I am prepared to accept calculated mean values to one decimal place only. Thus remove the second decimal value though out the manuscript.

b)     Similarly, please only report cashmere yield to one decimal place, as you have already multiplied the proportion by 100 to obtain percentage values.

c)     Similarly report CVDFD, bodyweight to one decimal.

L137 here and elsewhere, significant

L138, the value of -0.049 is not significant according to table 4.

Confusion, is it CV of diameter or CV of guard hair length?

Table 4, remove the empty row for BW.

Discussion

Please insert subheadings which will make it easier for the reader to understand the focus of the discussion.

L168, it is not cashmere production but cashmere processing.

L170-172. This statement is not supported by all studies of dehairing where fibre diameter and cashmere length are highly significant factors.

L176, fleece

L174-177. This does not accord with the results in Table 1, where the minimum hair length was 1 cm and the minimum for cashmere was 2.9 cm. This discussion needs to be very careful as it is assuming that the mid-side sample is suitable for hair length measurements. What about the hairs growing on the “edge” of the fleece near the belly, legs and back etc? Will they be long enough for dehairing? As the cashmere is actually combed, then what effect does combing actually have on guard hair length? This discussion should keep away from making generalised conclusions about dehairing given that further studies are required L178.

L178, replace trials with experiments.

L179, you mean the yields of the mid-side sample in this study compared with what type of samples elsewhere?

L179-181, rewrite or remove. Please specify if these yields for combed samples, or from mid-side samples or entire fleeces or bales of cashmere? This type of discussion is mis-leading and unhelpful.

It is not a competition as to who has the highest yielding cashmere especially as the measurements refer to very different types of greasy fibre. Given the SD of cashmere yield in the present study was 13%, it would appear that there are many goats in the study with a yield of less than 35% and many of less than 22% (that is the mean of 48% minus 2xSD). Is this correct? Perhaps discussing this variation in the population of these goats is more useful for both the readers and the processors than misleading comparisons about other breeds of goats.

L182-187, delete as not relevant.

L188-194, delete as not useful. Some of this may be helpful at the start of the discussion.

L195. This would be a better way to start the discussion.

L199, why would you do indirect selection for guard hair length when you could do direct selection for cashmere length? In any case most of the guard hairs remain on the goat after combing.

L203-204. If the correlation of -0.04 is not significant then say there was no significant correlation. It is not low and negative, it is not significant.

L200-206. How is this relevant if all the cashmere is combed? There is almost no guard hair in the combed fibre.

L207 -228. This section is confusing the authors. The weight are for small samples not the total fibre grown on the animals. It is logical that if you have taken a heavier sample then there will be more guard hair and likely more cashmere. If you have more guard hair it is likely to be either or both longer and coarser although you could have more total hairs for some goats, and so in a sample with lots of hair the cashmere yield will be lower (it is part of the calculation). So what is the real meaning of the correlation results for sample weights of hair and cashmere? I think that these results are confusing and not the main purpose of this preliminary investigation. It would be better to remove these sample weight correlation results and any discussion related to them so as to keep a clear focus on the results that may be informative. For example, in L228 correlations of -0.04 and -0.01 would not be significant and should not be presented as being converse evidence.

L230. The best measurement for cashmere homogeneity is fibre diameter SD, which has not been reported here. As the CV measurement is based on a division of SD divided by the mean, if the SD stays more or less stable then variation in the mean provides a variation in the CV. Is there any evidence that CV of fibre diameter affects cashmere processing or dehairing?

L237-240. This is speculation about another study. Delete the paragraph.

Conclusion

Given the severe limitations of the present study this conclusion is far too long.

Limitations to the study should be in the discussion.

There are more limitation to this study than listed here, specifically the assumption that the mid-side site is representative for both guard hairs and cashmere in the entire fleece and that guard hair length represents only one year of growth. Evidence in reference 29 indicated that guard hair length in Liaoning goats represented more than one year of growth. Given the variation in the dates of birth and in the dates of sampling it is highly likely that some goats have up to 18 months of hair growth (born January 2020 and sampled in June 2021) and others 12 months of hair growth (born May 2020 and sampled in May 2021). These hair lengths are then compared with 12 months of cashmere growth. This is a serious confounding of the hair length data.

More guard hairs by weight would increase the costs of transport, the scouring and dehairing. It appears that the authors need to think a lot more carefully regarding their conclusions.

L255, there is no evidence regarding price in this study so delete.

L277, there is no evidence supporting any selection strategy in this preliminary study so this line must be removed.

Author Response

Response to Reviewer 1

The authors wish to thank the anonymous reviewer for its valuable comments and constructive suggestions. The responses to points are provided below. We have tried to answer to all reviewer’s requests as well as possible. Important changes have been made according to reviewer suggestion.

Point 1: There is scientific interest in the development of the fleece of cashmere bearing goats and as the authors state, there are relatively few studies of the physical properties of the guard hair component. From this solid starting point the authors let themselves down by some poor methodology, poor understanding of scientific terms and confused reasoning.

There is a potential report here but it must have a more narrow focus, be very clear about the actual methods used, not be distracted by minor issues and avoid making conclusions about issues which are outside the narrow focus of the preliminary investigation. The discussion needs to focus on the main finding, if any, of the work. The data could potentially show that guard hairs are not useful for predicting cashmere content of a fleece compared with direct measurement of cashmere.

Response 1: We agree with the comments therefore important changes have been made to the paper, starting from the title in which we added “a preliminary study”. Our aim is not to use guard hair to predict the cashmere content but only to study the phenotypic relationship between the two types of fibre. We understand that some part of the study may be misleading therefore we provide a new version of the paper with important changes.

Point 2: The study has some important limitations which the authors do not seem to appreciate. They have not controlled for variation in birth and sampling dates which are highly likely to lead to differences in the period of growth of guard hair of up to 6 months and therefore confound the measurement of guard hair length. One of the cited references shows this type of result but is not used in the discussion.

Response 2: We agree with the comments. More limitation were described in the text according to the suggestion.

Point 3: authors also confuse themselves and the reader by not providing a clear explanation of the cashmere combing harvesting whereby most of the guard hair remains on the goat after the combing.

Response 3: It is true, however some guard hair are collected with the combing and for this reason the fibre is subjected to dehairing.

Point 4: There is also confusion by suggesting that an indirect method of assessing cashmere should involve measuring guard hair length, when it would be better to instead spend your time directly measuring cashmere length, especially given the low correlation between hair length and cashmere length in the present study (0.41), which explained only 16% of the variance in cashmere length. This result might indicate the opposite conclusion to the one presently made in the manuscript.

Response: We have changed some part of the discussion. Please see the new manuscript.

Point 5: There are already quite a few published papers on the usefulness of measuring cashmere length in predicting total cashmere production, but mid-side sampling is not necessarily a reliable method of predicting total cashmere production from cashmere goats (see later comments).

Response: We totally agree with the comment. We have removed any speculation on cashmere yield.

Specific comments

L68 replace few with little to read ... little information has been ...

Response: Change has been provided

L71 ... to it being a low value product.

Response: Change has been provided

L71, a main contributor to the lack of reporting is most likely the cost and time involved in measuring the hair diameter.

Response: This sentence has been added to the text.

L72, ... or possibly used to ...

Response: Change has been provided

L80 Do not start a sentence with a number. Remove the full stop after the number.

Response: The text has been changed

L92. Rewrite as awkward and potentially misleading expression.

Reference 27 is outdated and not suitable for cashmere and should be removed. Since 1968 more thorough studies with Merino sheep in Australia have shown that the mid-side site does not estimate the fibre diameter of wool tops as well as grid sampling of the fleece. McGregor 1994 also demonstrated systemic bias of mid-side sampling in cashmere goats for both cashmere content and also differences for cashmere fibre diameter. The real issue is that mid-side sampling provides reasonable ranking of goats if it is ranking that you desire but not necessarily predictions of cashmere fibre. This approximation will consequently modify any result and claims made about your subsequent results. What McGregor 1994 reported in relation to the mid-side site systematically overestimating the cashmere yield will impact upon the usefulness of guard hair measurements.

McGregor, B.A. (1994). Measuring cashmere content and quality of fleeces using whole fleece and midside samples and the influence of nutrition on the test method. Proceedings of the Australian Society of Animal Production 20: 186-189. This paper was easily found by Google on Research Gate.

Response: The sentence has been changed and the reference removed

L96-97. The sample would not be undisturbed if it had been washed in ether. Is the sequence of events correct?

L96. If the fibre sample was undisturbed then the length measurements would be the unstraightened lengths. Given that cashmere has natural waves which are called crimp, the actual cashmere length would be longer than those measured.

L96 and elsewhere. There seems to be something missing as the manuscript presents data for both the CV of guard hair length (lines 138, 140) and CV of cashmere length based on only one measurement per sample?

L98. You have not defined what you mean by staple weight or really how you selected a staple for weighing.

L99. To what precision were the weights recorded, 0.01, 0.001 g?

L100-101. There is confusion here, a proportion would be say 0.45, while the percentage would appear as 45%. Clarify this sentence.

L104. How were the snippets prepared?

Are you sure that 4000 snippets were measured on the hair samples without repeat measurements of the same field of view? As table 1 shows, some samples only had 0.02 g of hair to measure.

Response: We understand that any reference to weight of fibre and cashmere yield is inappropriate and inaccurate for the methodology used. In light of that we had removed the data from the manuscript along with any reference to the traits. We added that the measurement of the length was carried out on unstraightened fleece. We also had fixed the reference in the text to the “ CV of guard hair length and CV of cashmere length” with “CV of fibre diameter” with as it was a typing mistake. We added to the text that the snippets were prepared according to standard test method for diameter of wool and other animal fibers using an optical fiber diameter analyzer proposed by ASTM.

L109, replace carried on with undertaken.

Response: The text has been changed.

Guard hair length replace with Maximum guard hair length (see line 96);

Down length replace with Maximum crimped cashmere staple length.

Response: The text has been changed.

Table 1 and elsewhere in the manuscript.

  1. a) Please note that as the OFDA 100 only measures to the nearest 1 micrometre, I am prepared to accept calculated mean values to one decimal place only. Thus remove the second decimal value though out the manuscript.
  2. b) Similarly, please only report cashmere yield to one decimal place, as you have already multiplied the proportion by 100 to obtain percentage values.
  3. c) Similarly report CVDFD, bodyweight to one decimal.

Response: The text has been changed as suggested

L137 here and elsewhere, significant

Response: The text has been changed

L138, the value of -0.049 is not significant according to table 4. Confusion, is it CV of diameter or CV of guard hair length?

Response: The text has been changed

Table 4, remove the empty row for BW.

Response: The row was removed

L168, it is not cashmere production but cashmere processing.

Response: The word has been changed

L176, fleece

Response: The word has been changed

L174-177. This does not accord with the results in Table 1, where the minimum hair length was 1 cm and the minimum for cashmere was 2.9 cm. This discussion needs to be very careful as it is assuming that the mid-side sample is suitable for hair length measurements. What about the hairs growing on the “edge” of the fleece near the belly, legs and back etc? Will they be long enough for dehairing? As the cashmere is actually combed, then what effect does combing actually have on guard hair length? This discussion should keep away from making generalised conclusions about dehairing given that further studies are required L178.

Response: We agree with the observation. We changed the text using the conditional and stressing that this is a preliminary suggestion observed in a mid-side sample therefore the limitation is huge and more studies are required.

L179, you mean the yields of the mid-side sample in this study compared with what type of samples elsewhere?

L179-181, rewrite or remove. Please specify if these yields for combed samples, or from mid-side samples or entire fleeces or bales of cashmere? This type of discussion is mis-leading and unhelpful.

It is not a competition as to who has the highest yielding cashmere especially as the measurements refer to very different types of greasy fibre. Given the SD of cashmere yield in the present study was 13%, it would appear that there are many goats in the study with a yield of less than 35% and many of less than 22% (that is the mean of 48% minus 2xSD). Is this correct? Perhaps discussing this variation in the population of these goats is more useful for both the readers and the processors than misleading comparisons about other breeds of goats.

Response: As written above, we understand that any reference to weight of fibre and cashmere yield is inappropriate and inaccurate for the methodology used. In light of that we had removed the data from the manuscript along with any reference to the traits.

L182-187, delete as not relevant.

L188-194, delete as not useful. Some of this may be helpful at the start of the discussion.

Response: All these parts have been removed from the text.

L199, why would you do indirect selection for guard hair length when you could do direct selection for cashmere length? In any case most of the guard hairs remain on the goat after combing.

Response: This part has been removed from the text.

L203-204. If the correlation of -0.04 is not significant then say there was no significant correlation. It is not low and negative, it is not significant.

L200-206. How is this relevant if all the cashmere is combed? There is almost no guard hair in the combed fibre.

Response: This part has been removed from the text.

L207 -228. This section is confusing the authors. The weight are for small samples not the total fibre grown on the animals. It is logical that if you have taken a heavier sample then there will be more guard hair and likely more cashmere. If you have more guard hair it is likely to be either or both longer and coarser although you could have more total hairs for some goats, and so in a sample with lots of hair the cashmere yield will be lower (it is part of the calculation). So what is the real meaning of the correlation results for sample weights of hair and cashmere? I think that these results are confusing and not the main purpose of this preliminary investigation. It would be better to remove these sample weight correlation results and any discussion related to them so as to keep a clear focus on the results that may be informative. For example, in L228 correlations of -0.04 and -0.01 would not be significant and should not be presented as being converse evidence.

Response: All these parts have been removed from the text.

L230. The best measurement for cashmere homogeneity is fibre diameter SD, which has not been reported here. As the CV measurement is based on a division of SD divided by the mean, if the SD stays more or less stable then variation in the mean provides a variation in the CV. Is there any evidence that CV of fibre diameter affects cashmere processing or dehairing?

Response: We agree with the observation. However we focused on  CVDFD as to our knowledge this parameter is the one used in the textile industry to set the spinning of the cashmere.

L237-240. This is speculation about another study. Delete the paragraph.

Response: The paragraph has been removed

Limitations to the study should be in the discussion.

There are more limitation to this study than listed here, specifically the assumption that the mid-side site is representative for both guard hairs and cashmere in the entire fleece and that guard hair length represents only one year of growth. Evidence in reference 29 indicated that guard hair length in Liaoning goats represented more than one year of growth. Given the variation in the dates of birth and in the dates of sampling it is highly likely that some goats have up to 18 months of hair growth (born January 2020 and sampled in June 2021) and others 12 months of hair growth (born May 2020 and sampled in May 2021). These hair lengths are then compared with 12 months of cashmere growth. This is a serious confounding of the hair length data.

More guard hairs by weight would increase the costs of transport, the scouring and dehairing. It appears that the authors need to think a lot more carefully regarding their conclusions.

L255, there is no evidence regarding price in this study so delete.

L277, there is no evidence supporting any selection strategy in this preliminary study so this line must be removed.

Response: The conclusion paragraph has been strongly reduced, limitations have been improved and moved in the discussion section.

Reviewer 2 Report

The indicators examined in this manuscript are too simple to fully explain the phenotypic changes. It is suggested that authors add tests for phenotypic genes.

Author Response

Response to Reviewer 2

Point 1: The indicators examined in this manuscript are too simple to fully explain the phenotypic changes. It is suggested that authors add tests for phenotypic genes.

Response: The authors wish to thank the anonymous reviewer for its valuable comment and the time spent on reading the manuscript.

We've made many major changes to the manuscript, starting from the title. Our results come from a very preliminary study on phenotypic relationship between the upper and the down coat and this is outlined in the title and in the text, in particular in the limitations section.

We state that further studies are required in particular genetic studies. Any speculations on possible direct application of such results in a selection plan have been removed from the manuscript.

Reviewer 3 Report

This is a straight forward manuscript that clearly describes the correlation of parameters of down hair compared to guard hair in Cashmere goats.

I only have few comments:

- Line 203-204. Is there an explanation why there are converse result compared to Zhang et al.? This should be discussed.

- The manuscript should be edited by a native English speaker, but overall the manuscript is well written.

Author Response

Response to Reviewer 3

The authors wish to thank the anonymous reviewer for its valuable comment and the time spent on reading the manuscript.

This is a straight forward manuscript that clearly describes the correlation of parameters of down hair compared to guard hair in Cashmere goats.

I only have few comments:

Point 1: Line 203-204. Is there an explanation why there are converse result compared to Zhang et al.? This should be discussed.

Response: Thank you for the comment. As we've made many major changes to the manuscript, this part of the text has been removed as strongly recommended by reviewer 1: his point was that our sample is not comparable to the one used by Zhang and colleagues.

Point 2: The manuscript should be edited by a native English speaker, but overall the manuscript is well written.

Response: Thank you, however the text has been improved and checked by a British colleague.

Reviewer 4 Report

In general terms, it seems an interesting work to relate the length of the coarse fibers with the other quality characteristics and the productivity of the cashmere of a breed of goats from Inner Mongolia (P.R. China). However, the authors make an initial conceptual error when they express that these phenotypic measurements could encourage some consequence in the response to selection by them. It should be clarified that for this, in the end, some description of the genetic basis of the measured characteristics is required: heritability, genetic correlations, etc.

For the purpose of conceptually improving the article, any genetic connotation should be removed and only the measurements at the phenotypic level should be dealt with. Then, if genealogical data were available, the genetic description of the variables chosen in this work could be made. Just there, the response to the selection of each one of them could be inferred separately and the multiple responses if they were considered together.

On the other hand, the article is based on a small number of samples (n=158), from a population of an experimental station and only juvenile animals (up to one year). Adding to this a limited discussion of similar works does not envision a generalization of the conclusions.

For this reason, it is suggested to add to the discussion other works not taken into account by the authors and in order to obtain said generalization, at least in part. Suggested literature:

Castillo MF, Frank EN, Prieto A, et al. Development and validation of a commercial grade technique for Patagonian cashmere fiber. J Textile Eng Fashion Technol. 2022;8(2):54‒57. DOI: 10.15406/jteft.2022.08.00301

Redden H, Robson D, Rhind SM. Effect of a cashmere breeding program on fibre length traits. Aust J Agric Res. 2005;56:781–787.

Specific indications

Pag 1, lines 18-19: the fiber length after dehairing is the important trait. Although this is correlated with the length of crude fiber, other effects can modify the hauteur in the dehairing. Improve the wording please.

Page 2, lines 58-70: Since the ALBWCG population is extremely fine, a source should be included indicating up to what mean fiber diameter the thickening could be a textile problem. To better specify the problem, an average diameter of F1 Liaoning x ALBWCG should be included.

Page 2, lines 74-77: could this expression serve as a work hypothesis? , clarify, please. Only if that were the case, the word selection could be retained but only in this paragraph.

Page 2, lines 84-85: explain here at what age/live weight are kids marketed for meat?

Page 3, lines 96-99: please, explain better the procedure for measuring the length of the fiber, was the length measured in blocks or was a Baer diagram made? The accuracy of the analytical balance, please.

Page 3, lines 101-105: clarify whether only non-pigmented (white) fibers were measured or was there also colored fiber?

Pag 5, lines 168-173: if the de-hairing is so important, this part of the discussion is insufficient. List the characteristics and factors that affect dehairing and which ones could or could not be adequately found in this fiber. It is recommended to review the following literature:

Frank EN, Hick MVH, Castillo MF, et al. Determination of the optimal number of runs of dehairing in fibers of patagonian cashmere goats. J Textile Eng Fashion Technol. 2018;4(3):188‒190. DOI: 10.15406/jteft.2018.04.00144.

McGregor BA, Butler KL. The effects of cashmere attributes on the efficiency of dehairing and dehaired cashmere length. Textile Res J. 2008;78(6):486–96.

Page 5, lines 177-178: in this case it is absolutely necessary to carry out de-hairing tests, at least in the laboratory, in order to determine if this relationship is really a problem.

Page 5, lines 179-181: expand the references on yield. See the works:

Mcgregor BA. Scouring and Dehairing Australian Cashmere. Agri Futurres Goat Fibre. 2018. 34 p.

Frank EN, Hick MVH, Castillo MF, et al. Determination of the efficiency of the AM2 dehairing technology process with Llama fiber of different types of fleeces and Alpaca Huacaya fiber. J Textile Eng Fashion Technol. 2022;8(1):6‒8. DOI: 10.15406/jteft.2022.08.00293.

Page 6, lines 197-201: there is no genetic description, only phenotypic observation, to speculate on selection.

Author Response

Response to Reviewer 4

The authors wish to thank the anonymous reviewer for its valuable comment and the time spent on reading the manuscript. The responses to points are provided below. We have tried to answer to all reviewer’s requests as well as possible. Important changes have been made according to reviewer suggestion.

Point1: In general terms, it seems an interesting work to relate the length of the coarse fibers with the other quality characteristics and the productivity of the cashmere of a breed of goats from Inner Mongolia (P.R. China). However, the authors make an initial conceptual error when they express that these phenotypic measurements could encourage some consequence in the response to selection by them. It should be clarified that for this, in the end, some description of the genetic basis of the measured characteristics is required: heritability, genetic correlations, etc.

For the purpose of conceptually improving the article, any genetic connotation should be removed and only the measurements at the phenotypic level should be dealt with. Then, if genealogical data were available, the genetic description of the variables chosen in this work could be made. Just there, the response to the selection of each one of them could be inferred separately and the multiple responses if they were considered together.

Response 1: We agree with the comments therefore important changes have been made to the paper, starting from the title in which we added “a preliminary study”.

Our results come from a very preliminary study on phenotypic relationship between the upper and the down coat and this is outlined in the title and in the text, in particular in the limitations section.

We understand that some part of the study may be misleading therefore we provide a new version of the paper with important changes.

We state that further studies are required in particular genetic studies. Any speculations on possible direct application of such results in a selection plan have been removed from the manuscript.

Point 2: On the other hand, the article is based on a small number of samples (n=158), from a population of an experimental station and only juvenile animals (up to one year). Adding to this a limited discussion of similar works does not envision a generalization of the conclusions.

For this reason, it is suggested to add to the discussion other works not taken into account by the authors and in order to obtain said generalization, at least in part. Suggested literature:

Castillo MF, Frank EN, Prieto A, et al. Development and validation of a commercial grade technique for Patagonian cashmere fiber. J Textile Eng Fashion Technol. 2022;8(2):54‒57. DOI: 10.15406/jteft.2022.08.00301

Redden H, Robson D, Rhind SM. Effect of a cashmere breeding program on fibre length traits. Aust J Agric Res. 2005;56:781– 787.

Response 2: We agree with the comments. As written above the text was strongly reduced as suggested also by reviewer 1, in particular, the conclusion has been strongly reduce. We have removed any speculation on selection and we have improved the “limitation” section.

Specific indications

Pag 1, lines 18-19: the fiber length after dehairing is the important trait. Although this is correlated with the length of crude fiber, other effects can modify the hauteur in the dehairing. Improve the wording please.

Response: Thank you for the comment. However this section is the “simple summary” which is written for a general not specialist audience. We think it should be free from too may technical debates.

Page 2, lines 74-77: could this expression serve as a work hypothesis? , clarify, please. Only if that were the case, the word selection could be retained but only in this paragraph.

Response: Thank you for the comment. Any speculation on selection has been removed from the manuscript.

Page 2, lines 84-85: explain here at what age/live weight are kids marketed for meat?

Response: Unfortunately we don’t have this information.

Page 3, lines 96-99: please, explain better the procedure for measuring the length of the fiber, was the length measured in blocks or was a Baer diagram made? The accuracy of the analytical balance, please.

Response: We improved the description of the sample processing. Any reference to the analysis of the weight and cashmere yeald has been removed from the manuscript as strongly suggested by the reviewer 1.

Page 3, lines 101-105: clarify whether only non-pigmented (white) fibers were measured or was there also colored fiber?

Response: Thank you. This information has been added to the text.

Pag 5, lines 168-173: if the de-hairing is so important, this part of the discussion is insufficient. List the characteristics and factors that affect dehairing and which ones could or could not be adequately found in this fiber. It is recommended to review the following literature:

Frank EN, Hick MVH, Castillo MF, et al. Determination of the optimal number of runs of dehairing in fibers of patagonian cashmere goats. J Textile Eng Fashion Technol. 2018;4(3):188‒ 190. DOI: 10.15406/jteft.2018.04.00144.

McGregor BA, Butler KL. The effects of cashmere attributes on the efficiency of dehairing and dehaired cashmere length. Textile Res J. 2008;78(6):486–96.

Response: Thank you. Such references has been added.

Page 5, lines 177-178: in this case it is absolutely necessary to carry out de-hairing tests, at least in the laboratory, in order to determine if this relationship is really a problem.

Response: Thank you. We stated that in the text.

Page 5, lines 179-181: expand the references on yield. See the works:
Mcgregor BA. Scouring and Dehairing Australian Cashmere. Agri Futurres Goat Fibre. 2018. 34 p.

Frank EN, Hick MVH, Castillo MF, et al. Determination of the efficiency of the AM2 dehairing technology process with Llama fiber of different types of fleeces and Alpaca Huacaya fiber. J Textile Eng Fashion Technol. 2022;8(1):6‒8. DOI: 10.15406/jteft.2022.08.00293.

Response: Thank you for suggesting the references. However we have removed all the data on cashmere weight and yield as proposed by the reviewer one (our sample is inappropriate to any speculation on such parameters).

Page 6, lines 197-201: there is no genetic description, only phenotypic observation, to speculate on selection.

Response: We agree with you. Any speculation on selection has been removed from the paper.

Round 2

Reviewer 2 Report

The indicators examined in this manuscript are too simple to fully explain the phenotypic changes. It is suggested that authors add tests for phenotypic genes.

Author Response

(The authors gave the same response as above.)
